# Multimorbidity combinations, costs of hospital care and potentially preventable emergency admissions in England: A cohort study

**Jonathan Stokes**[1]*, **Bruce Guthrie**[2], **Stewart W. Mercer**[2], **Nigel Rice**[3], **Matt Sutton**[1]

**1** Centre for Primary Care and Health Services Research, University of Manchester, Manchester, United Kingdom, **2** Usher Institute, The University of Edinburgh, Edinburgh, Scotland, United Kingdom, **3** Department of Economics and Related Studies and Centre for Health Economics, University of York, York, United Kingdom

* jonathan.m.stokes@manchester.ac.uk

## Abstract

### Background

Patients with multimorbidities have the greatest healthcare needs and generate the highest expenditure in the health system. There is an increasing focus on identifying specific disease combinations for addressing poor outcomes. Existing research has identified a small number of prevalent "clusters" in the general population, but the limited number examined might oversimplify the problem and these may not be the ones associated with important outcomes. Combinations with the highest (potentially preventable) secondary care costs may reveal priority targets for intervention or prevention. We aimed to examine the potential of defining multimorbidity clusters for impacting secondary care costs.

### Methods and findings

We used national, Hospital Episode Statistics, data from all hospital admissions in England from 2017/2018 (cohort of over 8 million patients) and defined multimorbidity based on ICD-10 codes for 28 chronic conditions (we backfilled conditions from 2009/2010 to address potential undercoding). We identified the combinations of multimorbidity which contributed to the highest total current and previous 5-year costs of secondary care and costs of potentially preventable emergency hospital admissions in aggregate and per patient. We examined the distribution of costs across unique disease combinations to test the potential of the cluster approach for targeting interventions at high costs. We then estimated the overlap between the unique combinations to test potential of the cluster approach for targeting prevention of accumulated disease. We examined variability in the ranks and distributions across age (over/under 65) and deprivation (area level, deciles) subgroups and sensitivity to considering a smaller number of diseases.

There were 8,440,133 unique patients in our sample, over 4 million (53.1%) were female, and over 3 million (37.7%) were aged over 65 years. No clear "high cost" combinations of

**Data Availability Statement:** Data cannot be shared publicly because of a data sharing agreement. Data are available from NHS Digital (https://digital.nhs.uk/data-and-information/data-

tools-and-services/data-services/hospital-episode-statistics) for researchers who meet the criteria for access to confidential data.

**Funding:** The work is part of JS's personal research Fellowship, funded by the Medical Research Council (MRC Grant Ref: MR/T027517/1). The funders had no role in study design, data collection and analysis, decision to publish, or preparation of the manuscript.

**Competing interests:** The authors have declared that no competing interests exist.

**Abbreviations:** ACSC, ambulatory care sensitive conditions; HES, Hospital Episode Statistics; ICD-10, International Classification of Diseases, 10th revision; IMD, Index of Multiple Deprivation; NHS, National Health Service; RECORD, REporting of studies Conducted using Observational Routinely-collected Data.

multimorbidity emerged as possible targets for intervention. Over 2 million (31.6%) patients had 63,124 unique combinations of multimorbidity, each contributing a small fraction (maximum 3.2%) to current-year or 5-year secondary care costs. Highest total cost combinations tended to have fewer conditions (dyads/triads, most including hypertension) affecting a relatively large population. This contrasted with the combinations that generated the highest cost for individual patients, which were complex sets of many (6+) conditions affecting fewer persons. However, all combinations containing chronic kidney disease and hypertension, or diabetes and hypertension, made up a significant proportion of total secondary care costs, and all combinations containing chronic heart failure, chronic kidney disease, and hypertension had the highest proportion of preventable emergency admission costs, which might offer priority targets for prevention of disease accumulation. The results varied little between age and deprivation subgroups and sensitivity analyses.

Key limitations include availability of data only from hospitals and reliance on hospital coding of health conditions.

## Conclusions

Our findings indicate that there are no clear multimorbidity combinations for a cluster-targeted intervention approach to reduce secondary care costs. The role of risk-stratification and focus on individual high-cost patients with interventions is particularly questionable for this aim. However, if aetiology is favourable for preventing further disease, the cluster approach might be useful for targeting disease prevention efforts with potential for cost-savings in secondary care.

## Author summary

### Why was this study done?

- Multimorbidity, the presence of 2 or more chronic conditions in an individual, results in worse outcomes for patients and higher costs to health systems.

- Because existing interventions show little success, researchers have begun looking for specific combinations, "clusters," of conditions that might be treated more effectively.

- Existing analyses have focused on combinations of conditions that are common in the general population, but these may not be the combinations that are associated with the most important outcomes from a health service perspective.

### What did the researchers do and find?

- We used national hospital data to realign research into multimorbidity by starting with outcomes relevant to the healthcare system, total costs of secondary care, and costs of (potentially preventable) emergency admissions.

- We identified all unique combinations of 28 conditions present in a cohort of over 8 million people admitted to a hospital in England between April 2017 and March 2018.

- We examined the distribution and top contributors of total costs of these combinations over 1 and 5 years for the whole cohort and the average costs per individual.

- We additionally highlighted overlaps between unique combinations, which may merit follow-up in terms of development of further disease and costs.

- We identified over 60,000 unique disease combinations. No combination accounted for more than 3.2% of total costs for patients with multimorbidities. The combinations with the highest average cost per patient were not the same as those with highest total cost.

## What do these findings mean?

- There are no clear discrete disease combinations at which to target interventions, which implies a generalist/multidisciplinary team approach will remain important rather than pathways/guidelines based on a few specific disease clusters.

- Combinations containing the highest cost patients (the current focus of many interventions) were different to those accounting for the highest total costs, implying the need to develop interventions beyond only high-risk patients.

- There might be scope to use clusters to understand and develop preventative interventions, but focusing on addressing well-known disease risk factors (such as obesity, diet, exercise, and deprivation) with public health/primary care interventions might provide the most efficient route to benefit systems financially and benefit many patients with multimorbidities.

## Background

There are several outcomes that are important to health systems. However, especially in turbulent economic times, the costs (and cost-effectiveness) of care persistently rank high on the policy agenda. Particularly, the prevention of disease and expanded primary care are increasingly a global policy focus, aiming to reduce overall costs of comparatively expensive secondary care and potentially preventable emergency admissions [1,2]. These aims are not just important for the overall healthcare system, but also for individuals, as hospital admissions are not desirable in themselves and preventing illness at an early stage may avoid disability or future complex invasive treatment.

Patients with multimorbidities have the most contact with and generate the highest expenditure in the health system [3]. Multimorbidity is now a well-established priority for both research [4] and medical practice [5,6]. The accumulation of chronic conditions within an individual is associated with worse outcomes than having no chronic conditions or a single condition [7]. Multimorbidity is highly prevalent, occurring in roughly a third of community-based adults [8]. By aged 65 years, most people (65%) are multimorbid, and it is 2 to 3 times more common in the most socioeconomically deprived communities compared with the least deprived [9,10]. But, there has been little success to date in developing effective or cost-effective new models of care for these patients [11–13].

Given this lack of success, emphasis is now moving from simply describing "the problem of multimorbidity," by counting and documenting the number of conditions, towards potentially

more useful stratifications. This has taken the form of a focus on "disease clusters," examination of specific combinations of cooccurring conditions [4,5]. But, multimorbidity is a complex problem. If measuring, for example, the presence of 28 conditions, there are a theoretical ($2^{28}$ minus 29) >268 million unique combinations to consider.

## Identifying useful disease clusters

There are primarily 2 approaches in the current literature for simplifying multimorbidity to a manageable handful of subgroupings: (1) cluster analysis (grouping diseases); and (2) latent factor analysis (grouping patients) [14]. Both methods try to uncover hidden relationships between conditions using statistical methods. This research might prove to be useful if it manages to identify target clusters for direct intervention, or, via further research on aetiological mechanisms, target clusters for preventing disease accumulation. For example, for further examination within a cluster, whether: (1) one condition causes the other, (2) the conditions share common risk factor(s), (3) they share associated risk factors, or (4) the shared diagnostic features are actually due to another distinct condition [15].

## Methodological complexity

However, there is also a danger of oversimplification and impracticality in the output of these analyses. These complex statistical methods involve a number of modelling choices where adjustments to the technical implementation can give very different results [14]. There is likely to remain substantial within cluster variability [16]. Grouping conditions with cluster analysis then makes it difficult to relate to outcomes for individual patients, whereas grouping patients with latent factor analysis results in ultimately unobservable (i.e., latent) clusters since the same diseases can occur across multiple clusters [17]. There are further complex methods (with additional statistical assumptions) to try to assign individuals with a probability to the hidden latent factors [18], but it remains true that what we can actually observe clinically are symptoms, signs, and conditions.

## Clinical/Intervention complexity

Most importantly, not all clusters will be of equal significance in terms of the outcomes that policymakers want to address [19]. Cluster and latent class analyses currently concentrate on combinations most prevalent in the general population (with some latent factor studies subsequently measuring the average service use associated with the patients assigned to each cluster) [17]. However, it is not obvious whether highly prevalent clusters in the general population will be the same combinations associated with outcomes that place most pressure on the supply constraints of healthcare systems, such as costs of (potentially preventable) emergency admissions and overall costs of secondary care. Combinations with the highest secondary care costs may be priority targets for multimorbidity intervention and prevention, if there are indeed a manageable number with a large enough impact.

## Aims

National hospital data provide the opportunity to realign research into multimorbidity by starting with outcomes of relevance to the healthcare system rather than focusing on common sets observed in the general population. It also allows analysis of all observed disease combinations and their distribution rather than requiring selection of only a handful. We use national, administrative data from all National Health Service (NHS) hospital admissions in England in 2017/2018 to construct a cohort and examine the potential of defining multimorbidity clusters

for impacting secondary care costs (annual, over 1 year, 2017/2018, and previous 5 years, 2013/2014 to 2017/2018) via:

1. targeting interventions, examining: (i) if there are observable combinations of conditions contributing significantly to the secondary care costs/ "preventable admission" costs that can be targeted; (ii) if there are combinations of conditions with particularly high costs per patient that can be targeted.

2. targeting prevention of accumulated disease: (iii) combinations of conditions observed cross-sectionally might hide interrelationships that develop in the future through accumulation of further conditions. Examining whether the overlap between the unique combinations suggest conditions which merit follow-up in terms of development of multimorbidity.

## Methods

This study is not based on a specific prospective analysis plan. Analyses were planned in author discussions in January 2020. The study was reported in line with the REporting of studies Conducted using Observational Routinely-collected Data (RECORD) guidelines (see completed checklist, S1 Appendix) [20]. No specific ethics approval was required for this study.

### Context

The NHS is a tax-funded healthcare system with treatment provided free to patients at the point of delivery. NHS hospitals are predominantly paid by activity, according to a national tariff with Healthcare Resource Groups determining the prices for services [21]. NHS hospitals treat the vast majority of patients in England, with private providers accounting for, for example, just 1.3% of elective activity [22].

Our ability to record multimorbidity in this setting relies on good quality data recording multiple comorbidities for each patient. Accuracy of diagnosis recording has improved in the hospital setting in England since it was linked to payments in 2002. A systematic review in 2011 found median diagnostic recording accuracy of 80%, up to 96% for the primary condition [23]. Additionally, NHS guidance since 2010 has stated that "any co-morbidity that affects the management of the patient and contributes to an accurate clinical picture within the current episode of care must be recorded" and lists 61 comorbidities that must always be recorded regardless (15 of the 28 conditions we record in this study are on this list—see S2 Appendix) [24].

### Data

We use Hospital Episode Statistics (HES) data, an individual-level record of all contacts with NHS hospitals in England, from financial year, April to March, 2009/2010 to 2017/2018. Conditions are recorded using International Classification of Diseases (10th revision, ICD-10) coding. However, multiple comorbidities are only recorded in admission records, not visits to outpatient departments nor emergency department attendances. We restrict our sample to any patient with at least 1 elective or emergency admission in the most recent year, 2017/2018, ensuring the possibility of multimorbidity being recorded. We exclude maternity or other admission types, such as from high-security psychiatric hospitals (see S3 Appendix). We record the presence of 28 chronic conditions according to the coding reported in a previous study which identified validated algorithms for ICD-10 coding of 30 morbidities in administrative data for the study and surveillance of multimorbidity [25]. Of the 30, three conditions

were for types of cancer, which we combined into a single condition. The majority required only a single hospitalisation with a relevant ICD-10 code as any of the recorded conditions—exceptions requiring 2 hospitalisations were chronic pain, chronic viral hepatitis B, irritable bowel syndrome, and multiple sclerosis (see S2 Appendix). We create a separate identifier for each unique combination of these conditions present in the data.

We address any potential underrecording by using all previous data available for an individual patient, from 2009/2010, to backfill any missing diagnoses. Whenever a relevant ICD-10 code (one of the 28 chronic conditions) is observed, the condition "switches on" and is then carried forward over subsequent years. The final multimorbidity count for each patient, therefore, encompasses all accumulated conditions over the 2009/2010 to 2017/2018 period.

Our outcomes of interest are: (1) total costs of secondary care; and (2) costs of potentially preventable emergency admissions. We apply the national tariff costs to each of the inpatient, outpatient, and emergency department visits [21], and sum these to create total costs of secondary care. To examine potentially preventable costs, we separately calculate all of those costs directly attributable to an emergency inpatient admission with a primary diagnosis of an ambulatory care sensitive condition (ACSC, according to the definition used by Harrison and colleagues, see S4 Appendix) [26]. Five-year costs are additionally summed for each patient over 2013/2014 to 2017/2018, again based on the hospital activity data linked to the applicable national tariff for each year.

## Statistical analysis

We first describe the sample, comparing across number of chronic conditions to exemplify the outcomes across counts and then by each individual condition recorded. We then aggregate the patient level dataset to the level of each unique combination of multimorbid conditions.

## Targeting interventions

We rank the combinations on each outcome (total secondary care costs/ACSC costs) and then examine the distribution and focus in more detail on the top 10 unique combinations that attract the highest hospital costs. This is done overall and separately for costs per patient by dividing by the count of patients contributing to that cost.

## Targeting prevention of accumulated disease

We examine the number of overlapping combinations also containing the conditions in the top 10 ranked and present a total cost for all related combinations using a bubble plot, a scatter plot allowing a third dimension represented by the size of the circle. We weight the circle size by proportion of potentially preventable (ACSC) costs.

## Subgroup analysis

To examine heterogeneity of the findings, we repeat the above stratifying by age (over/under 65) and area deprivation (using deciles of the Index of Multiple Deprivation, IMD) [27]. We focus on rank for these subgroups, since population-level estimates for the denominator (i.e., number of multimorbid persons in total) are not routinely available. Those with missing age or deprivation are excluded from the subgroup analyses but included in the main analysis which does not require these variables.

### Sensitivity analysis

Following peer-review feedback, we were asked to explore the effects of dropping hypertension as a condition, as arguably it might be considered a symptom or risk-factor, and to consider the effects of only including a smaller number of conditions on the interpretation of results. We added these as sensitivity analyses, rerunning the main analysis exploring the distribution of costs and top 10 conditions contributing to total secondary care costs for all unique: (1) combinations of 27 conditions, excluding hypertension from our original list; and (2) combinations of 15 conditions, the 15 conditions with explicit NHS guidance on comorbidity recording (as described above and in S2 Appendix).

## Results

### Overall sample

Overall, there were 8,440,133 unique patients in our sample with an elective or emergency inpatient admission in 2017/2018 (roughly 15% of the total population of England) [28]. Over 4 million (53.1%) were female, and over 3 million (37.7%) were aged over 65 years. Nearly 5 million (57.6%) had at least 1 chronic condition, and over 2 million (31.6%) had multimorbidity (2 or more of the 28 recorded chronic conditions). The patients with multimorbidities had a higher average age and IMD score (more deprived), increasing with number of conditions as expected. Costs increased across all categories with number of conditions, average total cost of secondary care for those with 4+ conditions was 5.2 times as much as patients with no chronic conditions, and costs of potentially preventable emergency admissions 20.5 times as much (see Table 1).

Overall, patients with no chronic conditions (42.4% of the sample) contributed to 23.3% of the total secondary care costs, patients with 1 chronic condition (25.8% of the sample) to 21.4% of the total costs, patients with multimorbidities (31.8% of the sample) to 55.3% of the total costs.

Table 2 shows the sample characteristics by individual condition. Hypertension was the most prevalent morbidity recorded in the sample, recorded in over a quarter of patients (26.5%). Diabetes (11.6%), chronic kidney disease (10.3%), and asthma (9.5%) were next most common, recorded in a tenth of the sample. In terms of costs, patients with hypertension contributed to 41.3% of total costs of secondary care, followed by chronic kidney disease (24.3%), both higher than the total contribution of those with no conditions.

### Targeting interventions

We found 63,124 unique combinations of conditions for patients with multimorbidities in 2017/2018 (out of a total theoretical of >268 million).

Only 7 of the 63,124 unique combinations of co-conditions contribute any more than 1% of total cost of secondary care for patients with multimorbidities, with a maximum of approximately 3.2% in any 1 unique set (Table 3). Only 2 unique combinations contribute more than 1% of total cost of secondary care for all patients, max of 1.7%. This is the same for ACSC costs and for 5-year costs (see S5 Appendix). So, many different multimorbidity combinations each account for a very small fraction of each outcome.

The combinations with the highest total costs of secondary care in 2017/2018 are the same as those with the highest cost over 5 years. There is also a lot of overlap with the rank of combinations with the highest potentially preventable (ACSC) costs. Hypertension is common to all of the highest cost combinations. The majority of combinations are dyads (with 2 triads at ranks 4 and 9).

**Table 1. Sample characteristics by number of chronic conditions recorded.**

| | Number of conditions | | | | |
|---|---|---|---|---|---|
| | 0 | 1 | 2 | 3 | 4+ |
| Percent of sample (n) | 42.4% (3,577,253) | 25.8% (2,180,649) | 15.4% (1,303,279) | 8.6% (729,548) | 7.7% (649,404) |
| Age, mean (SD) | 39.1 (24.7) | 54.3 (21.9) | 64.4 (18.1) | 69.7 (15.8) | 73.8 (13.9) |
| MISSING | 73,247 | 50,592 | 40,543 | 26,648 | 25,795 |
| Aged over 65 years | | | | | |
| No, % (n) | 82.1% (2,877,930) | 62.3% (1,326,834) | 43.7% (552,078) | 32.2% (226,649) | 22.9% (142,678) |
| Yes, % (n) | 17.9% (626,076) | 37.7% (803,223) | 56.3% (710,658) | 67.8% (476,251) | 77.1% (480,931) |
| MISSING | 73,247 | 50,592 | 40,543 | 26,648 | 25,795 |
| Sex | | | | | |
| Male, % (n) | 47.3% (1,692,306) | 46.6% (1,015,926) | 46.8% (609,703) | 46.5% (339,463) | 46.1% (299,442) |
| Female, % (n) | 52.7% (1,883,567) | 53.4% (1,164,155) | 53.2% (693,419) | 53.5% (390,034) | 53.9% (349,949) |
| MISSING | 1,380 | 568 | 157 | 51 | 13 |
| Area deprivation, mean (SD) | 22.3 (15.9) | 22.0 (15.8) | 22.5 (15.9) | 23.2 (16.2) | 24.2 (16.4) |
| MISSING | 119,693 | 82,174 | 55,373 | 33,230 | 29,905 |
| MM count, mean (SD) | 0.0 | 1.0 | 2.0 | 3.0 | 4.7 (1.0) |
| Total cost of secondary care 2017/2018 (£), mean (SD) | 1,708.5 (2,499.2) | 2,577.0 (3,472.6) | 3,716.9 (4,646.6) | 5,276.0 (5,821.3) | 8,922.2 (8,212.7) |
| Inpatient cost (£), mean (SD) | 1,320.9 (2,363.5) | 2,022.7 (3,259.8) | 3,027.6 (4,384.0) | 4,444.7 (5,518.5) | 7,830.5 (7,753.2) |
| A&E cost (£), mean (SD) | 92.1 (151.5) | 116.9 (197.2) | 154.6 (249.5) | 208.9 (307.3) | 342.9 (458.4) |
| Outpatient cost (£), mean (SD) | 295.5 (388.5) | 437.5 (566.7) | 534.7 (715.7) | 622.4 (818.9) | 748.8 (999.7) |
| ACSC cost (£), mean (SD) | 47.7 (339.4) | 125.4 (666.3) | 227.7 (1078.9) | 410.5 (1515.5) | 977.8 (2638.7) |
| Total cost of secondary care (£, 5 years), mean (SD) | 4,069.0 (6184.9) | 6,733.5 (8573.2) | 9,453.2 (10,840.2) | 12,786.4 (22,675.7) | 19,728.3 (17,543.3) |
| Inpatient cost (£, 5 years), mean (SD) | 2,926.3 (5,402.8) | 4,969.3 (7,498.8) | 7,209.1 (9,568.7) | 10,041.1 (21,850.9) | 16,131.4 (15,460.7) |
| A&E cost (£, 5 years), mean (SD) | 234.6 (357.6) | 315.7 (520.2) | 404.5 (675.1) | 527.2 (831.8) | 824.6 (1278.4) |
| Outpatient cost (£, 5 years), mean (SD) | 908.1 (1267.2) | 1,448.5 (1,760.6) | 1,839.7 (2,239.8) | 2,218.1 (2,663.4) | 2,772.3 (3,280.6) |
| ACSC cost (£, 5 years), mean (SD) | 121.3 (686.1) | 337.9 (1,506.1) | 600.4 (2,193.8) | 1,018.1 (18,822.9) | 2,116.2 (4,811.2) |

ACSC, ambulatory care sensitive conditions, potentially preventable emergency admissions; A&E, Accident & Emergency department; MM, multimorbidities.

This is in sharp contrast to the highest cost combinations for any individual patient, combinations which have many conditions (high complexity, frequently including a mental health condition, chronic pain, diabetes, or chronic pulmonary disease) but only affect 1 or 2 patients overall (Table 4).

## Targeting prevention of accumulated disease

Fig 1 examines the overlap of the top 10 combinations with other unique combinations containing the same conditions. All combinations containing both chronic kidney disease + hypertension contribute nearly a third (29.1%) of total costs of secondary care for patients with multimorbidities (16.1% of costs for all patients). Combinations containing diabetes + hypertension contribute nearly a quarter (24.3%) of total costs of secondary care for patients with multimorbidities (13.5% of costs for all patients). The combinations containing chronic heart failure + hypertension, and chronic heart failure + chronic kidney disease + hypertension each have the highest proportion of total cost that is from potentially preventable admissions (both 18.0%).

## Subgroup analysis

As expected, the over 65s and most deprived subgroups exhibited the most complexity, a larger number of unique combinations of conditions compared to under 65s and the wealthiest.

**Table 2. Sample characteristics by morbidity.**

| Morbidity | Sample prevalence (%) | Proportion of total secondary care costs in 2017/2018 (%) |
|---|---|---|
| Alcohol misuse | 3.0 | 4.42 |
| Asthma | 9.5 | 11.02 |
| Atrial fibrillation | 0.9 | 2.12 |
| Cancer | 5.4 | 12.2 |
| Chronic heart failure | 4.4 | 12.1 |
| Chronic kidney disease | 10.3 | 24.3 |
| Chronic pain | 8.1 | 12.8 |
| Chronic pulmonary disease | 7.0 | 13.6 |
| Chronic viral hepatitis B | 0.8 | 1.4 |
| Cirrhosis | 0.8 | 2.4 |
| Dementia | 3.2 | 6.8 |
| Depression | 6.7 | 10 |
| Diabetes | 11.6 | 18.8 |
| Epilepsy | 1.9 | 3.3 |
| Hypertension | 26.5 | 41.3 |
| Hypothyroidism | 4.9 | 7.9 |
| Inflammatory bowel disease | 1.7 | 2.1 |
| Irritable bowel syndrome | 1.2 | 1.6 |
| Multiple sclerosis | 0.5 | 0.7 |
| Myocardial infarction | 1.2 | 3.1 |
| Parkinson disease | 0.7 | 1.5 |
| Peptic ulcer disease | 0.6 | 1.1 |
| Peripheral vascular disease | 0.0 | 0 |
| Psoriasis | 0.4 | 0.7 |
| Rheumatoid arthritis | 2.4 | 4.3 |
| Schizophrenia | 0.6 | 0.8 |
| Severe constipation | 2.9 | 7.5 |
| Stroke or TIA | 1.4 | 3.9 |
| No condition | 42.4 | 23.3 |

NB: Columns are not exclusive so do not sum to 100%.

However, the distributions and top ranked combinations of conditions contributing to each outcome were highly consistent across subgroups. The most deprived and under 65s also featured combinations including depression and alcohol misuse in the top 10 (see S6 Appendix).

## Sensitivity analysis

Both sensitivity analyses gave broadly the same results as above, no clear cost driving combinations, and combinations with highest cost those with fewer conditions but affecting a larger proportion of the population. When dropping hypertension, the highest cost combination was chronic kidney disease and diabetes, contributing to 2.4% of total secondary care costs for patients with multimorbidities, 1.1% of costs for all patients (see S7 Appendix). When analysing only 15 conditions, there were by design far fewer unique combinations, 5,869 observed instead of 63,124. This inflated the proportions of total costs slightly, with more of the population assigned to each category. The highest cost combination was, like the main results, diabetes and hypertension, now contributing to 6.5% of total secondary care costs for patients with multimorbidities, but still only 2.9% of costs for all patients (see S8 Appendix).

**Table 3. Top 10 ranked unique multimorbidity combinations contributing to total costs of secondary care for patients with multimorbidities in 2017/2018.**

| Rank | Conditions in combination (count) | Percent of total cost for MM patients (%) | Percent of total cost for all patients (%) | Total cost of secondary care (£m) | Cost of ACSC emergency admissions (£m; rank) | Total 5-year cost of secondary care (£m; rank) | Count of unique patients with combination (% of all MM patients) |
|---|---|---|---|---|---|---|---|
| 1. | Diabetes, hypertension (2) | 3.16 | 1.74 | £457.23 | £36.89 (1) | £1,304.99 (1) | 171,420 (6.39%) |
| 2. | Kidney, hypertension (2) | 2.38 | 1.31 | £344.15 | £14.76 (7) | £832.85 (2) | 76,324 (2.85%) |
| 3. | Cancer, hypertension (2) | 1.68 | 0.93 | £242.76 | £2.86 (50) | £518.58 (4) | 45,479 (1.70%) |
| 4. | Kidney, diabetes, hypertension (3) | 1.56 | 0.86 | £226.61 | £18.94 (6) | £611.65 (3) | 42,753 (1.59%) |
| 5. | Pulmonary, hypertension (2) | 1.21 | 0.67 | £174.78 | £22.72 (3) | £498.82 (5) | 53,062 (1.98%) |
| 6. | Pain, hypertension (2) | 1.20 | 0.66 | £173.23 | £4.41 (31) | £472.20 (6) | 54,792 (2.04%) |
| 7. | Asthma, hypertension (2) | 1.12 | 0.62 | £162.79 | £9.41 (13) | £449.75 (7) | 63,263 (2.36%) |
| 8. | Hypertension, hypothyroidism (2) | 0.83 | 0.46 | £119.98 | £4.51 (30) | £295.78 (8) | 38,540 (1.44%) |
| 9. | CHF, CKD, hypertension (3) | 0.80 | 0.44 | £115.26 | £20.22 (4) | £255.10 (10) | 15,332 (0.57%) |
| 10. | CHF, hypertension (2) | 0.73 | 0.40 | £105.74 | £19.89 (5) | £258.21 (9) | 22,174 (0.83%) |

ACSC, ambulatory care sensitive conditions; CHF, chronic heart failure; CKD, chronic kidney disease; MM, multimorbidities.

## Discussion

### Summary of results

We analysed a sample of over 8 million patients with any inpatient admission in 2017/2018 using national data in England. There are no clear multimorbidity combinations for a targeted cluster intervention approach to reduce secondary care costs significantly for the health system. Aiming interventions at the highest cost patients alone appears to be particularly questionable, since complex combinations which were highest cost for individual patients were negligible in terms of total system spend. However, all overlapping combinations containing chronic kidney disease + hypertension, or diabetes + hypertension, made up a significant proportion of total secondary care costs for patients with multimorbidities. Combinations containing chronic heart failure + chronic kidney disease + hypertension had the highest proportion of total cost that is from potentially preventable emergency admissions, which might offer relative priority targets if accumulation is preventable after further examination of aetiology.

### Findings in context

Early evidence has shown that there is a curvilinear, near-exponential, relationship between disease count and costs [29], and multimorbidity can better explain variation in health and social care costs than age [30]. We illustrate the differences in costs of all observable unique combinations of conditions and highlight the differences in importance of combinations contributing high costs at the population versus individual level.

Similar to previous research examining potential to reduce costs of secondary care [31,32], our findings indicate the misconceptions of a focus on interventions targeting the highest-cost

**Table 4. Top 10 ranked unique multimorbidity combinations contributing to average annual costs of secondary care for any individual patient with multimorbidity in 2017/2018.**

| Rank | Conditions in combination (count) | Percent of total cost for MM patients (%) | Average annual cost of secondary care per patient (£) | Count of unique patients with combination (% of all MM patients) | Percent of total cost potentially preventable (%) |
|---|---|---|---|---|---|
| 1. | Alcohol misuse, asthma, hep B, cirrhosis, depression, diabetes, IBS, ulcer, stroke (9) | 0.002 | £333,011.00 | 1 (<0.01%) | 0 |
| 2. | Asthma, cancer, CHF, CKD, pain, pulmonary, depression, diabetes, hypertension, hypothyroidism, IBS, Parkinson's, arthritis, schizophrenia, constipation, stroke (16) | 0.002 | £258,950.00 | 1 (<0.01%) | 5.14 |
| 3. | CKD, pulmonary, hypertension, hypothyroidism, Parkinson's, severe constipation (6) | 0.001 | £143,166.00 | 1 (<0.01%) | 0 |
| 4. | CKD, chronic pain, pulmonary, diabetes, IBS, arthritis (6) | 0.001 | £139,508.00 | 1 (<0.01%) | 0 |
| 5. | Alcohol misuse, CHF, chronic pain, pulmonary, depression, diabetes, epilepsy, hypertension, psoriasis, severe constipation (10) | 0.001 | £135,218.00 | 1 (<0.01%) | 28.90 |
| 6. | Cancer, CHF, chronic pain, hypothyroidism, IBS, schizophrenia (6) | 0.001 | £122,212.00 | 1 (<0.01%) | 0 |
| 7. | Asthma, atrial fibrillation, CHF, CKD, chronic pain, pulmonary, depression, diabetes, hypertension, MI, psoriasis, arthritis, stroke (13) | 0.001 | £121,986.00 | 1 (<0.01%) | 19.38 |
| 8. | Asthma, atrial fibrillation, CHF, CKD, chronic pain, pulmonary, depression, diabetes, hypertension (8) | 0.002 | £119,983.50 | 2 (<0.01%) | 8.60 |
| 9. | Alcohol misuse, chronic pain, pulmonary, depression, epilepsy, MS, arthritis, severe constipation (8) | 0.001 | £117,937.00 | 1 (<0.01%) | 0.85 |
| 10. | CKD, pulmonary, diabetes, hypothyroidism, IBS, severe constipation (6) | 0.001 | £113,131.00 | 1 (<0.01%) | 3.62 |

CHF, chronic heart failure; CKD, chronic kidney disease; hep B, chronic viral hepatitis B; IBS, irritable bowel syndrome; MI, myocardial infarction; MM, multimorbidities; MS, multiple sclerosis; pulmonary, chronic pulmonary disease.

patients alone. The majority of system costs come from the bulk of patients with relatively fewer complex combinations of conditions indicating that the greatest effect might be made by reducing risk factors for the wider population.

The most recent latent class analysis of multimorbidity clusters in England analysed a sample of 113,211 patients with multimorbidities in primary care. The study identified 20 patient clusters across 4 age strata, calculating mean number of hospitalisations for patients in each cluster. The 2 highest utilising clusters were described according to the 3 conditions estimated to be most distinctive in the cluster: (1) chronic pain (81% of patients in cluster), coronary heart disease (53%), and depression (45%), with an average of 1.6 hospital spells per year; (2) coronary heart disease (61%), atrial fibrillation (53%), and chronic heart failure (49%), with an average of 1.5 hospital spells per year. Over 70% of patients in each of these clusters also had hypertension [17]. We build on this previous literature by analysing all unique, clinically observable combinations and highlight those most important in terms of secondary care costs. The focus on distribution in terms of outcome is also vital to understand the possible applications of a clustering approach.

## Strengths and limitations

The study incorporates data from a large national cohort of over 8 million hospitalised patients, all admitted patients in England in 2017/2018, and examines costs of hospital utilisation by all observed combinations of 28 common chronic conditions.

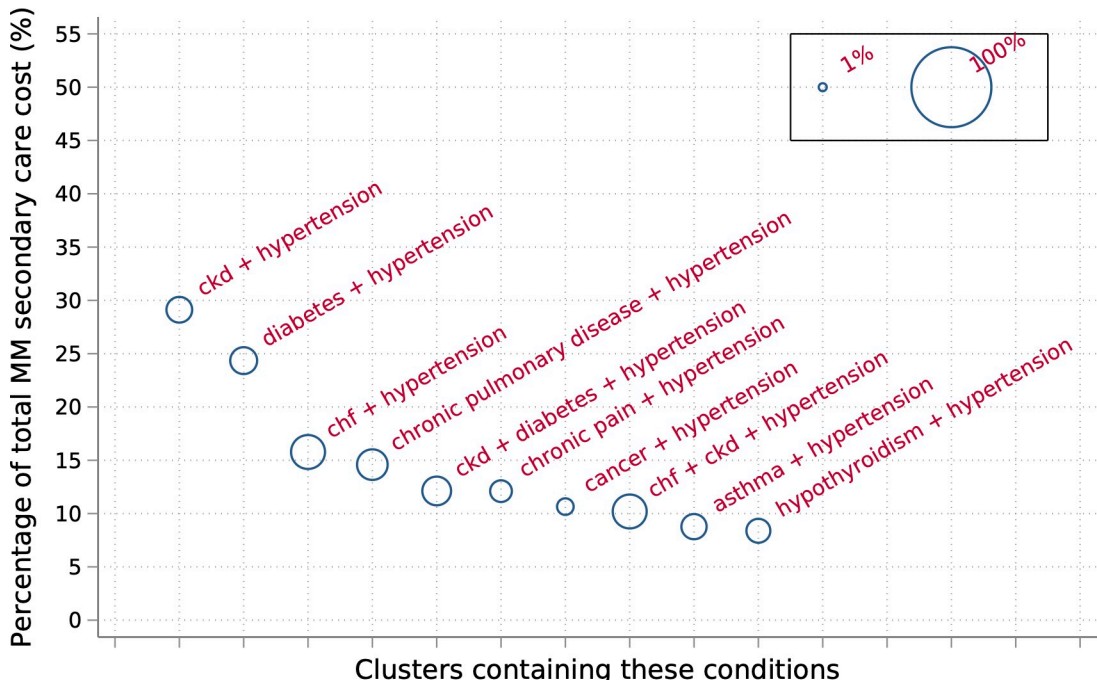

NB: ckd = chronic kidney disease; chf = chronic heart failure

**Fig 1. Bubble plot showing total contribution to costs of secondary care for patients with MM, summing costs of all patients with multimorbidities who have (at least) those conditions.** Proportion of total costs of secondary care for patients with multimorbidities shown on the y-axis. Area of circles are weighted by proportion of total cost that is potentially preventable (ACSC), larger circle indicates higher proportion of potentially preventable costs. ACSC, ambulatory care sensitive conditions; chf, chronic heart failure; ckd, chronic kidney disease; MM, multimorbidities.

However, we are only able to analyse a population with a hospital admission, which is likely to be different than a multimorbid population receiving other care services. Nevertheless, this is the costliest care setting and likely to contain the most complex patients. It also includes elective, including day cases, as well as emergency admissions totalling 15% of the total population of England. Accordingly, this would appear to be the most relevant population for our research question. We also observed only a fraction of the 268 million possible condition combinations, so there are likely to be many other drivers of costs in other patients in other years. However, we do not expect the distribution of costs or costliest combinations to change drastically with the inclusion of more rare combinations.

Without data from other settings, we are also only able to record costs in the hospital setting. Costs in other settings are likely to be highly relevant, particularly social care, primary care, and care home (long-term care) costs, and condition combinations are likely to exhibit different cost structures across different settings. However, arguably we are focusing on the most relevant setting in terms of NHS spending, since social care and care homes are largely paid privately by individuals with a proportion paid publicly by Local Authorities, and primary care costs on a predominantly per capita basis.

Unfortunately, there is no commonly agreed definition or list of ICD-10 codes for multimorbidity research [4], but we reproduce coding reported in a previously published methods paper which attempts to correct this [25]. Fifteen of the 28 conditions we focus on appear in the list of conditions the NHS specifically requires providers to record in this setting. Nevertheless, an audit conducted in 2014 identified consistent underrecording of comorbidities, particularly for those whose recording is subject to clinical decision on relevancy [33]. We use all

available patient data over multiple years to attempt to fill any missing data, in line with recommendations from analysis of administrative data in other settings [34], and sensitivity analysis to check the impact of more compulsory coding. Additionally, two of the conditions (chronic pain and cancer) do not appear in the compulsory NHS recording list but appear in our list of top 10 cost conditions suggesting they are being recorded well. However, this backfilling of condition codes does not allow us to incorporate the dynamic nature of multimorbidity, i.e., that new conditions can develop over time, which means those coded as multimorbid might not have been multimorbid for the entire period (particularly the previous 5-year cost analysis).

The definition of potentially preventable (ACSC) emergency admissions is not perfect, either. Eight of the 28 conditions we count overlap directly with the list of ACSC admissions meaning people with combinations containing these conditions have a higher probability of having an ACSC admission. However, this definition of potentially preventable admission is widely used and comparable with other studies.

### Implications for policy and practice

Our results indicate no clear target combination(s) of diseases for intervention in terms of impacts on secondary care costs. The plurality likely indicates that a generalist or multidisciplinary team approach to management will remain important rather than pathways/guidelines based on specific clusters.

We identified large differences in combinations contributing to high costs for the system (whole population) compared to high costs for an individual patient. Many of the current models of care attempt to target high-cost individuals with risk stratification tools and case management [12]. Developing models of care with benefits beyond high-risk patients is important.

The individual conditions contributing to the top costs overall are already known to be linked to risk factors, such as obesity, diet, exercise, and deprivation [35,36]. Focusing on addressing these well-known risk factors with public health/primary care interventions might provide a route to benefit systems financially and benefit many patients with multimorbidities [37]. However, this would likely involve increased funding/staffing in other parts of the system to deliver, so overall system savings are not guaranteed. There is some evidence that preventative interventions, particularly public health, can be highly cost-effective, nevertheless [38–40].

### Future research

Examining accumulation of conditions and ordering will be important in future studies, preferably using primary care or linked data. Examining the costs for combinations of multimorbidity across other settings would also be valuable and possible with other data sources. The effects of current payment tariffs on multimorbidity costs and differences across condition combinations also merits further work.

### Conclusions

There are no clear multimorbidity combinations for a cluster-targeted intervention approach to reduce overall secondary care costs. The role of risk stratification and focus on individual high-cost patients with interventions is particularly questionable for this aim. However, if aetiology is favourable for preventing further disease, the cluster approach might be useful for targeting disease prevention efforts with potential for cost-savings in secondary care. Given the individual conditions, a focus on reducing well-established risk factors in the general population is also likely to be beneficial for many patients with multimorbidities and might be more cost-effective.

## Supporting information

**S1 Appendix. The RECORD statement.** Checklist of items, extended from the STROBE statement, that should be reported in observational studies using routinely collected health data. (DOCX)

**S2 Appendix. Multimorbidity ICD-10 codes.**
(DOCX)

**S3 Appendix. Cohort sample coding.**
(DOCX)

**S4 Appendix. Ambulatory care sensitive conditions—emergency admissions.**
(DOCX)

**S5 Appendix. Distribution of costs per unique multimorbidity (MM) combination.**
(DOCX)

**S6 Appendix. Top 10 combinations by age and deprivation.**
(DOCX)

**S7 Appendix. Distribution and top 10 combinations, dropping hypertension.**
(DOCX)

**S8 Appendix. Distribution and top 10 combinations, only 15 NHS guidance conditions.**
(DOCX)

## Author Contributions

**Conceptualization:** Jonathan Stokes, Bruce Guthrie, Stewart W. Mercer, Nigel Rice, Matt Sutton.

**Data curation:** Jonathan Stokes.

**Formal analysis:** Jonathan Stokes.

**Funding acquisition:** Jonathan Stokes.

**Investigation:** Jonathan Stokes.

**Methodology:** Jonathan Stokes, Nigel Rice, Matt Sutton.

**Project administration:** Jonathan Stokes.

**Writing – original draft:** Jonathan Stokes.

**Writing – review & editing:** Jonathan Stokes, Bruce Guthrie, Stewart W. Mercer, Nigel Rice, Matt Sutton.

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
