## [Editor Report · Decision Letter 0]

3 Sep 2020

Dear Dr Stokes, 

Thank you for submitting your manuscript entitled "Multimorbidity combinations, costs of hospital care and potentially preventable emergency admissions: Cross-sectional analysis of a national dataset of over 8 million people" for consideration by PLOS Medicine.

Your manuscript has now been evaluated by the PLOS Medicine editorial staff and I am writing to let you know that we would like to send your submission out for external peer review.

Kind regards,

Helen Howard, for Clare Stone PhD 

Acting Editor-in-Chief

PLOS Medicine 

plosmedicine.org

---

## [Decision Letter · Decision Letter 1]

30 Sep 2020

Dear Dr. Stokes,

Thank you very much for submitting your manuscript "Multimorbidity combinations, costs of hospital care and potentially preventable emergency admissions: Cross-sectional analysis of a national dataset of over 8 million people" (PMEDICINE-D-20-04285R1) for consideration at PLOS Medicine. 

Your paper was evaluated by a senior editor and discussed among all the editors here. It was also evaluated by three independent reviewers, including a statistical reviewer (r#1). The reviews are appended at the bottom of this email and any accompanying reviewer attachments can be seen via the link below:

[LINK]

In light of these reviews, I am afraid that we will not be able to accept the manuscript for publication in the journal in its current form, but we would like to consider a revised version that addresses the reviewers' and editors' comments. Obviously we cannot make any decision about publication until we have seen the revised manuscript and your response, and we plan to seek re-review by one or more of the reviewers. 

We expect to receive your revised manuscript by Oct 21 2020 11:59PM. Please email us (plosmedicine@plos.org) if you have any questions or concerns.

We look forward to receiving your revised manuscript. 

Sincerely,

Emma Veitch, PhD

PLOS Medicine

On behalf of Adya Misra, PhD, Senior Editor, 

PLOS Medicine

plosmedicine.org

*At this stage, we ask that you include a short, non-technical Author Summary of your research to make findings accessible to a wide audience that includes both scientists and non-scientists. The Author Summary should immediately follow the Abstract in your revised manuscript. This text is subject to editorial change and should be distinct from the scientific abstract. Please see our author guidelines for more information: https://journals.plos.org/plosmedicine/s/revising-your-manuscript#loc-author-summary

*In the last sentence of the Abstract Methods and Findings section, please include a brief note about any key limitation(s) of the study's methodology.

*Please clarify if the analytical approach reported here corresponds to one laid out in a prospective protocol or analysis plan? Please state this (either way) early in the Methods section.

*It would be helpful to use an appropriate reporting guideline to support reporting of the study, and perhaps also to trigger filling in some of the key details noted by reviewers (below). The RECORD guideline (https://www.equator-network.org/reporting-guidelines/record/) may be most appropriate here, designed to support reporting of studies done using routinely-collected health data. If using this guideline please note that in the methods section and include a completed checklist as a supporting information file (which should then be called out in Methods). 

Comments from the reviewers:

Reviewer #1: I confine my remarks to statistical aspects of this paper.

I have one question: Were the circles in the figure proportional by the area or the diameter of the circle? (It should be area). 

But this is a very interesting problem, statistically speaking. As the authors note, with so many conditions, there are millions and millions of possible combinations, far more than even the large number of subjects. The authors look at every combination that appeared in the data, no matter how complex. This runs the risk of overfitting the data. Although the authors did not do regression or something similar, they did look at the cost per patient of some very complex combinations. But it seems likely that many of the combinations that were not observed would have very high costs, too. 

I think a couple of additional analyses would be very useful.

First, what if all the costs per patient were randomly assigned in some fashion? 

Second, the authors could look at all unique dyads and triads of conditions. With 28 conditions there are 28*27/2 = 378 dyads and 9828 triads. This suits the sample size. 

Third, perhaps some of the conditions could be combined? This is more a medical question than a statistical one, so I can't really suggest combinations. But, as an educated layperson, it seems to me that there should be some combination possible. With 10 "meta-conditions" there would be 2^10 -11 - 1013 combinations and this is much more manageable. Even 15 conditions wouldn't be too bad. 

Peter Flom

Reviewer #2: REVIEW of "Multimorbidity combinations, costs of hospital care and potentially preventable emergency admissions: Cross-sectional analysis of a national dataset of over 8 million people".

The article provides interesting insights into the usefulness, limits, and potential of the cluster analysis approach to healthcare usage, has a very large sample. However, the paper need further improvement efforts to be published:

-Title

Please state that your study's population is not general national population, but hospitalized patients population

-Methods

Please give some context information about UK NHS 

If you collected outcome into five years, this is not a cross sectional design but a retrospective cohort design

Please explain how do you count for the five years costs and which time period do you analyze.

Please provide a unique subheading "statistical analysis" grouping the actual subheadings "analysis", "targeting intervention", "targeting prevention"….

First phrase of methods could be moved into the discussion.

The part of analyses, which you define as sensitivity analyses, could be better defined as analyses of subgroups, please revised in the text

-Results

There's no mentioning of the prevalence of the most common conditions among your population sample. I suggest you to provide a table one with the description of sample characteristics and the prevalence of the most common conditions among your sample (regardless of combinations) to understand how it is representative on entire population. 

You can create a figure with two columns representing in one the distribution of number of conditions in the sample (e.g. 42,4% no condition…) and in other the distribution of total costs by condition group (23.3% of cost for no condition)

In table 1 is lacking sometime of labels of the outputs (for example age will be reported mean and standard deviation). Moreover, please provide percentages out of bracket and absolute numbers inside. Please provide both currencies reported in the labels. Why there are so many with missing age it not compulsory in health care records to add date of birth? 

In table 2 do you report only in pounds? You refer the labels of columns as "total costs", but do they refer to "average total costs"?

In table 1 and 2 please provide how frequent ( prevalence of combinations) are the combinations in the MM patients

Please give at least one decimal in the figure of percentage reported in the text

Fig.1: The chart's meaning is a bit difficult to understand at first sight. I suggest you introduce it with a more descriptive title, and elaborate its description a bit further in the main text. Also, I'd move the NB note above, as part of the figure's description. Finally, the chart's key placement is unfortunate: I suggest you move it on the top right corner and frame it, in order for the reader not to confuse it with the actual data displayed by the figure.

-Discussion

Please move the subheading "findings in the context" before "limits"

The prescribing expenditure/drugs is not on per capita basis

You state "….so overall system saving are by no means guarantee….." this doubt is verified in the cost effectiveness analyses, which consider costs for each health care level. There are some evidences that found that prevention interventions even if they increase primary care costs but reducing other costs, produce net savings.

-Conclusion 

"There are no clear multimorbidity combinations for a targeted intervention approach to reduce secondary care costs. The role of risk-stratification and focus on individual high-cost patients is particularly questionable." The link between these two phrases isn't very apparent. I suggest you add proper logical connectors, or re-arrange the paragraph to make it sound more consequential.

Reviewer #3: Summary

The goal of this paper is to estimate clusters of comorbidities and examine these clusters with high health care costs and avoidable hospitalizations using health administrative databases. The question and approach are interesting; however, there needs to be additional detail added to clarify the methods and outcomes. 

Comments

1. I found the points in the introduction very interesting, but they could be organized so that it is easier for the reader. I would suggest separating the analytic problems form the clinical/intervention gaps. So first clearly explain why the clusters are hard to action from an intervention perspective, and second why the methods used to identify these clusters may be limited. This study then aims to address both those gaps.

2. The time element of the questions was not defined. The authors pose very interesting questions regarding the relationship between different conditions with health system outcomes (costs and avoidable hospitalization), but when? Within a year? Within 30 days? It wasn't until the midpoint of the methods that it seems like it cost over five years - but unclear form when (from the first diagnosis?). The period for when the hospitalization and costs accrual relative an index date could be more precise and justified.

3. Justification for the 28 conditions should be included, more than just the reference to the previous study (i.e. why were those the conditions chosen in the last study).

4. The validity of the definitions used for the 28 conditions is unclear. Many of the administrative data definitions should be validated, and those should be mentioned. Although the authors cite diagnostic accuracy, it's unclear if they are referring to only one coding or where in the diagnostic field the code should occur (i.e. most responsible diagnosis or can be a secondary field). Generally, multiple codes/visits are needed, but it was unclear if this was done. 

5. Interesting that 15 conditions are given explicit guidance on coding given the NHS guidance. This, however, may introduce bias in that these conditions are more likely to be coded and therefore conditions not on this list may be missed. Can the authors comment on this?

6. Hypertension is very dominant, and many argue not a condition but rather a risk factor for a chronic condition (such as diabetes or CHF). If hypertension was not included, the results would indeed be dramatically different. Can the authors comment on if this?

7. I agree overall with the interpretation that focusing on specific combinations is not likely to yield a specific path; however, I think that this does not necessarily lead to a generalist approach mentioned in the Discussion. Instead, multidisciplinary health care teams may be an equally beneficial approach - given a different mix of specialists depending on the patient's needs.

[LINK]

---

## [Decision Letter · Decision Letter 2]

4 Dec 2020

Dear Dr. Stokes,

Thank you very much for re-submitting your manuscript "Multimorbidity combinations, costs of hospital care and potentially preventable emergency admissions: Retrospective cohort analysis of a national dataset of over 8 million hospitalised people" (PMEDICINE-D-20-04285R2) for review by PLOS Medicine.

I have discussed the paper with my colleagues and the academic editor and it was also seen again by xxx reviewers. I am pleased to say that provided the remaining editorial and production issues are dealt with we are planning to accept the paper for publication in the journal.

[LINK]

We look forward to receiving the revised manuscript by Dec 11 2020 11:59PM. 

Sincerely,

Adya Misra, PhD

Senior Editor 

PLOS Medicine

plosmedicine.org

Requests from Editors:

Title: please consider revising to “Multimorbidity combinations, costs of hospital care and potentially preventable emergency admissions in England: A cohort study”

Abstract- “Multimorbid patients generate the highest expenditure in the health system and..” can we rephrase this to be less stigmatising?

Abstract- in the background section, could you please highlight the aim of your study

Abstract-please provide brief participant demographics. In addition "aged 65 years" in the abstract and results

Abstract “Given the individual conditions, a focus on reducing well established

risk factors in the general population is also likely to be beneficial for many

multimorbid patients and might be more cost-effective” was this directly tested? I suggest removing

 "In this study, we noted that ..." or similar at the start of the final subsection of the abstract

Page 5 – “multimorbidity is highly prevalent” could you add brief numbers to put this into context. In addition, I suggest you specify what you mean by “older” and “deprived” here.

Could you please rewrite the aims so they are not posed as questions

Methods- perhaps the mention of Beveridge is not needed? 

Methods- Please specify that specifics ethics approval was not needed for this study

RECORD checklist- please add paragraph numbers and refer to the completed checklist at the start of the methods section

Please can you revise all iterations of “multimorbid patients” to “patients with multimorbidities”

I suggest revising this sentence on page 15 “Aiming interventions to

target combinations with the highest cost patients alone appears to be particularly questionable” to highlight what your findings suggest.

I suggest removing lifestyle on page 16 “lifestyle risk factors, such as obesity, diet, exercise and deprivation”.

Please ensure all acronyms are introduced on first view

I suggest adding “strengths” to the limitations section on page 15-16

Please remove funding information from the main text

Please remove spaces from the square brackets

Comments from Reviewers:

Reviewer #1: The authors have addressed my concerns and I now recommend publication

Reviewer #2: The paper has been greatly improved and the authors modified the text accordantly to previous remarks

Reviewer #3: Thank you for taking the time to carefully respond to the comments. The revisions very much improve the clarity of the manuscript and the additional sensitivity analyses - especially those regarding hypertension - do provide a strong assurance of the robustness and significance of the findings.

[LINK]

---

## [Editor Report · Decision Letter 3]

5 Jan 2021

Dear Dr. Stokes,

I am writing concerning your manuscript submitted to PLOS Medicine, entitled “Multimorbidity combinations, costs of hospital care and potentially preventable emergency admissions in England: A cohort study.”

We have now completed our final technical checks and have approved your submission for publication. You will shortly receive a letter of formal acceptance from the editor.

Kind regards,

PLOS Medicine